# Exploring the Association of HLA Genetic Risk Burden on Thalamic and Hippocampal Atrophy in Multiple Sclerosis Patients

**DOI:** 10.3390/genes13112136

**Published:** 2022-11-17

**Authors:** Silvia Santoro, Ferdinando Clarelli, Paolo Preziosa, Loredana Storelli, Miryam Cannizzaro, Elisabetta Mascia, Federica Esposito, Maria Assunta Rocca, Massimo Filippi

**Affiliations:** 1Laboratory of Human Genetics of Neurological Disorders, Institute of Experimental Neurology, IRCCS San Raffaele Scientific Institute, 20132 Milan, Italy; 2Unit of Neurology, IRCCS San Raffaele Scientific Institute, 20132 Milan, Italy; 3Neuroimaging Research Unit, Institute of Experimental Neurology, Division of Neuroscience, IRCCS San Raffaele Scientific Institute, 20132 Milan, Italy; 4Vita-Salute San Raffaele University, 20132 Milan, Italy; 5Neurophysiology Service, IRCCS San Raffaele Scientific Institute, 20132 Milan, Italy

**Keywords:** multiple sclerosis, brain atrophy, HLA, genetic risk score

## Abstract

Multiple sclerosis (MS) is a complex disease of the central nervous system for which human leukocyte antigen (HLA) alleles are major contributors to susceptibility. Several investigations have focused on the relationship between HLA and clinical parameters, while few studies have evaluated its correlation with brain magnetic resonance imaging (MRI) measures. We investigated the association between the HLA genetic burden (HLAGB), originating from the most updated HLA alleles associated with MS, and neuroimaging endophenotypes, with a specific focus on brain atrophy metrics. A monocentric Italian cohort of 334 MS patients with imputed HLA alleles and cross-sectional volumetric measures of white matter (WM), gray matter (GM), hippocampus, thalamus and T2-hyperintense lesions was investigated. Linear regression models with covariate adjustment were fitted for each metric. We detected no effect of HLAGB on WM and GM volumes. Interestingly, we found a marginal correlation between higher HLAGB and lower hippocampal volume (β = −0.142, *p* = 0.063) and a nominal association between higher HLAGB and lower thalamic volume (β = −0.299, *p* = 0.047). No association was found with T2 lesion volumes. The putative impact of higher HLAGB on hippocampus and thalamus suggests, if replicated in independent cohorts, a possible cumulative contribution of HLA risk loci on brain volumetric traits linked to clinical deficits in MS.

## 1. Introduction

Multiple sclerosis (MS) is a chronic autoimmune disease of the central nervous system characterized by inflammation and neurodegeneration. It is a complex and clinically heterogeneous disorder, whose etiology arises from an interplay between genetic and environmental factors. It has been known for several decades that the major histocompatibility complex (MHC), also known as human leukocyte antigen (HLA) in humans, located on the short arm of chromosome 6 and encoding for cell-surface proteins and molecules responsible for the regulation of the immune system, harbors the strongest contributor to MS risk, the HLA-DRB1*15:01 allele [1], and the most known protective locus for MS, the HLA-A*02:01 allele [2]. Further fine mapping of associated loci revealed multiple independent susceptibility signals in the MHC region [2,3]. A recent large-scale genomic screen performed by the International Multiple Sclerosis Genetics Consortium (IMSGC) on >100,000 subjects revealed a markedly polygenic architecture for MS susceptibility, finally identifying 233 genome-wide significant loci, of which 32 were within the MHC region [4]. Beside contribution to MS susceptibility, an important question concerns the impact of HLA susceptibility loci on disease expression, in particular on MRI outcomes, which are a privileged set of measures to assess inflammation and neurodegeneration in white matter (WM) and gray matter (GM). In particular, neuroimaging metrics of brain atrophy are sensitive markers for capturing the neurodegenerative processes that better correlate with the accumulation of disability and cognitive impairment in MS [5,6]. More specifically, the atrophy processes involving deep GM regions are important biological substrates of clinical disability and cognitive impairment in MS. Indeed, significant neuropathological and neuroimaging studies indicate that thalamic and hippocampal involvement are important contributors to cognitive and mental disability, memory impairment and depression [7,8]. Recently, a multicenter study confirmed that the atrophy of hippocampal and deep GM nuclei are key factors associated with cognitive impairment in MS [9].

Regarding the MHC region, a few studies investigated its contribution to MRI phenotypes in multiple sclerosis, analyzing single loci [10,11,12] or the cumulative HLA genetic burden (HLAGB) [13,14]. Recently, a focus on the association between the complexity of MHC genetic architecture and MRI brain phenotypes has been elucidated in the context of the huge cohort of healthy volunteers enrolled in the UK Biobank [15]. Interestingly, 15 HLA alleles have been found to be significantly associated with at least one brain MRI outcome (*p* < 5 × 10^−8^) [15].

In the present study, we report the influence of the HLAGB score, derived from the most updated panel of MS-associated HLA alleles, on measures of brain atrophy (WM, GM, hippocampus and thalamus volumes) and on T2 lesion volume (T2LV), evaluated with cross-sectional MRI scans in an Italian cohort of MS patients. In addition, taking advantage of the aforementioned recent paper by Bian and colleagues [15], we explored the impact of the published HLA alleles on MRI outcomes in our MS Italian cohort.

## 2. Materials and Methods

### 2.1. Enrolled Patients

A cohort of 407 MS patients of Italian origin was enrolled at the MS Center in San Raffaele Hospital (HSR). Among them, 276 subjects had a relapsing remitting (RRMS) course, 72 had secondary progressive MS (SPMS) and 59 had primary progressive MS (PPMS).

Subjects were considered eligible for the study if: (i) their age at first MRI scan was ≥18 and ≤55 years and (ii) at least one MRI scan was available for the following metrics: white matter volume (WMV), gray matter volume (GMV), hippocampal volume (HippV), thalamic volume (ThalV) and T2LV. In case of longitudinal scans, the closest to diagnosis was selected.

The acquisition, segmentation and analysis of brain MRI images were performed as previously described [16].

### 2.2. Genotyping and Imputation

Genotype data were assayed with the Infinium OmniExpress (San Diego, CA, USA) (n = 253), Infinium Omni2.5 (n = 28) and Human610Quad (n = 53) IIlumina^®^ platforms (San Diego, CA, USA), with the application of standard filters to remove samples and SNPs of low quality. SNPs located in the MHC region (chromosome 6p21.3, from 29 Mb to 34 Mb) were extracted, and classical HLA class I and II alleles, SNPs and amino-acid polymorphisms were separately imputed from each platform array using SNP2HLA [17] and subsequently merged. The reference panel was derived from the Type 1 Diabetes Genetics Consortium and consisted of 5225 subjects of European ancestry, genotyped on 8926 common SNPs spanning the MHC region. After imputation, we retained loci with a high-quality imputation score (r^2^ > 0.6).

### 2.3. Selected HLA Loci

We used the most updated set of MS-associated HLA loci (n = 32) derived from the largest multi-center study [4]. The HLAGB was calculated for each individual as the sum of imputed HLA associated alleles, weighted by the natural log of odds ratios as previously reported in Appendix A in the original published paper from IMSGC [4], aligning alleles so that their effect corresponded to increased risk.

We further selected the 15 HLA alleles robustly associated with the MRI-based phenotypes at *p* < 5 × 10^−8^ that emerged from the UK Biobank; they have been tested in a healthy population and are available in Appendix A in the original published article [15].

### 2.4. Statistical Analysis

The association of HLAGB with MRI metrics was investigated by fitting linear regression models, adjusting for suitable covariates. Due to skewness to the right of T2LV distribution, its values were transformed on a natural log scale to approach normality.

We considered gender, age and disease duration at MRI scan, disease course (RRMS/PMS), being under treatment (y/n) and the chip used in HLA imputation as candidate covariates to be incorporated in the models. This set of variables was screened with a bootstrap resampling procedure [18] as implemented in bootStepAIC R package, to identify, for each MRI metric, the optimal set of independent covariates and to avoid the inclusion of spurious noise variables. We performed a forward stepwise selection across 1000 boostrapped samples, and all predictors that were identified upon minimization of Akaike Information Criterion (AIC) in at least 60% of the bootstrap runs were included in the final model.

We next evaluated the impact of the 29 single loci on the five neuroimaging outcomes, fitting linear models under minor allele additive coding, with the same set of covariates elicited for the HLAGB. Finally, for the two major contributors to MS risk, the HLADRB1*15:01 risk allele and the HLA-A*02:01 protective allele, we investigated the joint impact of the two loci by classifying patients as “high risk” (homozygote presence of the risk allele and homozygote absence of the protective allele, allowing for at most one heterozygote) or “low risk” (homozygote absence of the risk allele and homozygote presence of the protective allele, allowing for at most one heterozygote).

The present study was approved by the ethics committee in HSR (protocol number: 107/INT/2018), and written informed consent was obtained from the patients before study enrollment.

## 3. Results

Starting from the initial cohort of 407 enrolled subjects, we selected 334 patients with age at MRI scan between 18 and 55 years, whose clinico-demographic and neuroimaging metrics are summarized in Table 1. A total of 81.7% of the patients were under treatment at MRI scan, with details of therapies reported in Table 1.

As for HLA susceptibility loci, we excluded two of them because of lower imputation quality (r^2^ < 0.6), whereas the SNP rs114071505 was not present in the reference panel. A final set of 29 HLA loci was thus included in the score, with a mean imputation quality of 0.97 (range: 0.76–1). Regarding the MHC alleles emerged from the UK Biobank study, 10 out of 15 signals were available in our imputed HLA data as further reported.

With our sample size, we estimated a power of 73.3% and 94.3% at a significance level of 0.01 to account for multiple testing, assuming a variance explained by HLAGB of 3% and 5%, respectively.

Given the importance of the age predictor for brain atrophy measures, this variable was always retained in bootstrapped samples for the detection of stable covariates; age and disease duration were moderately correlated (r = 0.478) but with negligible multi-collinearity, as estimated by the variance inflation factor (VIF_Age_ = 1.391, VIF_DD_ = 1.302).

In the evaluation of the cumulative influence of the 29 HLA loci on atrophy metrics, adjusted models revealed a marginal correlation between higher HLAGB and decreased volume of hippocampus (β = −0.142, *p* = 0.063) and a nominally significant association with lower thalamic volume (β = −0.299, *p* = 0.047), although not upon multiple testing correction. No association was detected between the HLAGB and T2LV.

Association statistics between HLAGB and the five brain imaging outcomes are reported in Table 2, along with the outcome-specific included covariates.

The effect of the HLA genetic risk score on thalamic volume is shown in Figure 1.

When considering the single-locus analyses, we could not detect significant associations of any of the 29 loci with the investigated MRI outcomes after multiple testing correction: the overall pattern of association is illustrated in Figure 2, whereas details of association statistics are reported in Appendix A.

As regarding the major contributors to MS risk in MHC region (HLA-DRB1*15:01 risk allele and HLA-A*02:01 protective allele), after the stratification of patients belonging to the entire cohort in low- (n = 76) and high- (n = 50) risk groups, we found no evidence of association with any of the investigated metrics.

From the single-locus analyses of the 10 HLA alleles from Bian et al. [15], we could not detect any significant associations with MRI outcomes after multiple testing correction. Details of the association statistics are reported in Appendix A.

We further conducted association analysis of HLAGB score with the five MRI metrics upon stratification for disease course. For this purpose, we investigated the effect of the cumulative score in patients with a relapsing–remitting course (n = 253) and those with a progressive course, aggregating SPMS (n = 49) and PPMS (n = 32). We observed a nominal correlation between higher HLAGB and decreased thalamic volume (β= −0.673, *p*= 0.053) in the progressive group (Table 3), whereas no significant correlation was observed in the RRMS group.

## 4. Discussion

Despite the fundamental advancements in the knowledge of MS genetic component provided by large-scale international efforts, less is known on the genetic determinants impacting disease expression. The influence of established MS risk loci on disease phenotypes is debated, in particular with regard to surrogates of disease activity and severity such as MRI outcomes, and the literature is scattered with conflicting results. Indeed, a study by Isobe et al. [13] investigating in a cohort of 586 MS patients the cumulative impact of HLA risk loci on clinical and MRI outcomes revealed a sex-specific association of HLAGB with subcortical GM fraction in women, mostly driven by the HLA-DRB1*15:01 haplotype. These findings were nevertheless not replicated in a subsequent study by Mulhau et al. [19] in a similarly sized German cohort of patients. More recently, a study by Smets et al. [14] again could not find evidence of an association of HLA genetic risk score with MS phenotypes like the magnetization transfer ratio and volumetric measures.

In this study, we investigated the impact of HLAGB on cross-sectional MRI atrophy measures, taking advantage of a well characterized monocentric cohort of 334 Italian MS patients: in line with previous studies, we did not identify significant correlation with GMV and WMV on the entire cohort and, upon stratification on gender, we did not observe an association in females. We showed no evidence of association between HLAGB and T2LV.

A distinguishing feature of our study was, however, the availability of volumetric measures for important deep GM regions such as thalamus and hippocampus: indeed, our data provided some support for a putative impact of higher HLA risk score with lower thalamic volume, although this association did not withstand multiple testing correction.

Of interest, after stratification on disease course, this association was also observed in the progressive patients, whereas in the RRMS group the correlation was not significant, despite the evidence of the same direction of effect. This finding suggests that the signal emerged in the entire cohort is probably mostly driven by the progressive group.

It is known that MHC-I molecules play a critical role in the development, synaptic connectivity and plasticity of the CNS [20]. Furthermore, MHC-I proteins are particularly expressed in the neurons of the lateral geniculate nucleus of the thalamus, and it is also demonstrated that MHC class I is involved in the axonal and neurite outgrowth of hippocampal neurons in vitro [21].

The thalamus is a central relay structure for different cortical–subcortical circuits underpinning a broad range of functions as sensorimotor, cerebellar, visual, and cognitive [22]. Thalamic atrophy, potentially induced by the indirect degeneration secondary to the compromised projecting tracts from the WM, is a distinctive feature of all MS clinical phenotypes, contributing to explaining clinical disability and cognitive impairment [22,23].

As regarding single-HLA-locus analysis, we did not detect an association of the major contributor to the MS risk HLA-DRB1*15:01 allele with investigated MRI measures. This is in contrast with findings from previous studies, which found an association of the presence of this allele with a higher T2 lesion volume and to a declined volume of the normalized brain in MS patients [10]. Nevertheless, it should be mentioned that, in more recent studies, investigators found no clear effect of the HLA-DRB*1501 allele on gray matter fraction and cortical lesion volume [24], as well as no effect on brain parenchymal fraction and gray matter fraction [25].

Interestingly, the lack of replication in our cohort of the 10 single loci associated with MRI phenotypes in European unrelated subjects could be related to the lower sample size of our cohort as compared to the around 30 k available in UK Biobank. Secondly, given that some immune-related variants impact specific brain structure and that some of them have been previously implicated in brain disorders, we can speculate a different mechanism in MS, as an example of autoimmune brain disorder, in which the immune component has a different impact than in unaffected subjects.

A potential limitation of our investigation is that we did not impose a restriction for concomitant exposure to treatment at MRI scan, a limitation that is, however, shared with other studies; nevertheless, in our stability procedure for the detection of covariates, we could not identify a relevant impact of this variable on the investigated outcomes.

## 5. Conclusions

Concluding, our investigation confirmed previous findings stating no clear effect of HLA genetic score on WM and GM atrophy. Likewise, no evidence of an association with HLAGB was found with T2LV as a marker of disease activity. It is of course possible that other loci in the MHC region, different from those involved in disease risk, may affect the processes underlying neurodegeneration and atrophy. However, we found a nominal association, not investigated before, between HLAGB and the volume of the thalamus. After stratification on disease course, this association was also observed in progressive patients. If replicated in larger independent cohorts, this finding could interestingly point to a possible contribution of HLA risk loci on brain GM phenotypes linked to neurological deficits in MS patients.

## Figures and Tables

**Figure 1 genes-13-02136-f001:**
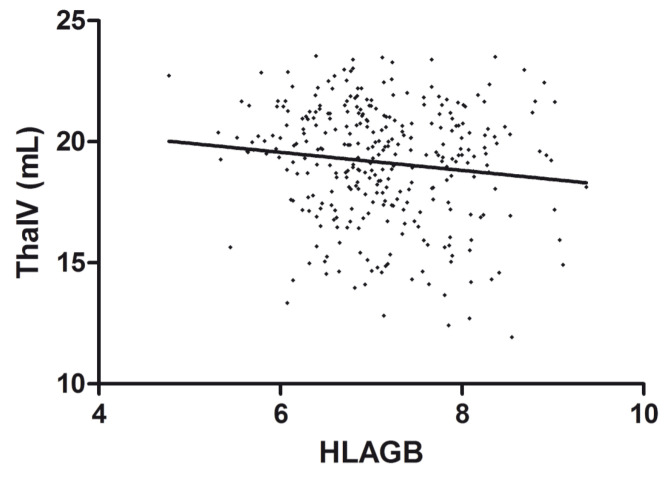
Correlation between HLAGB and thalamic volume. Scatter plot of the HLAGB and thalamic volume (ThalV) with an overlaid regression line. Each dot corresponds to a patient.

**Figure 2 genes-13-02136-f002:**
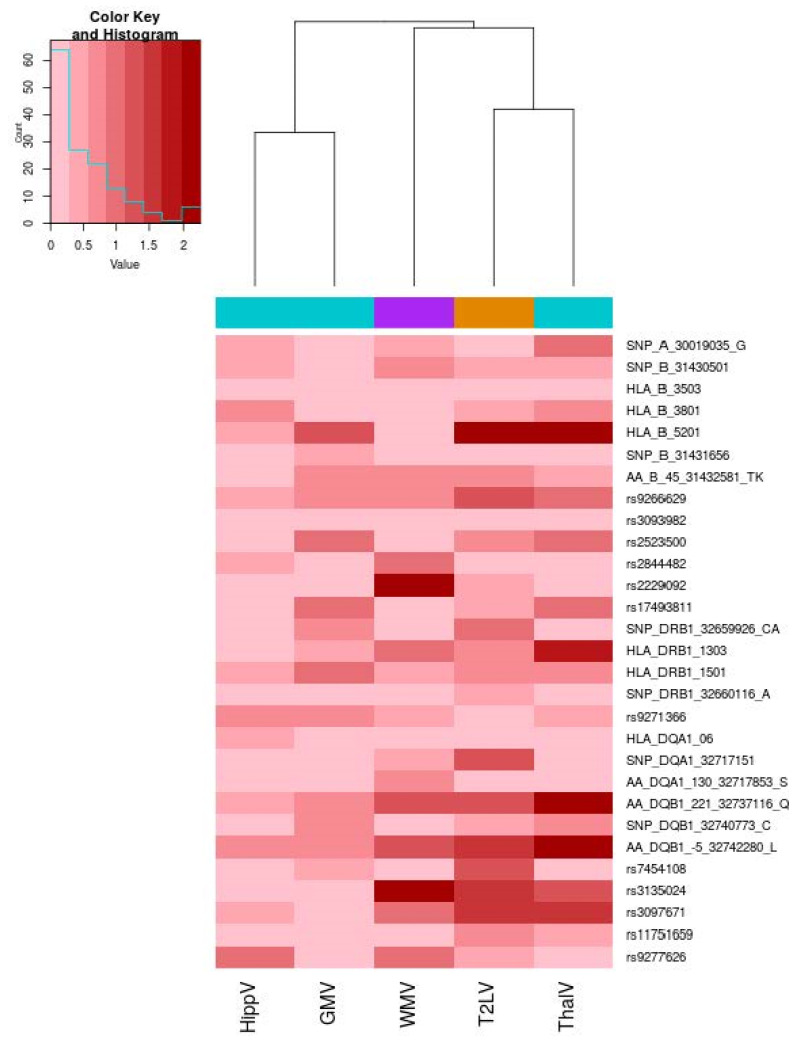
Heatmap of the association of 29 HLA loci and MRI outcomes. The heatmap illustrates the association level for the 29 investigated HLA MS susceptibility loci. The color map from pink to dark red represents the level of association in terms of −log10 (*p*). On the right, each HLA locus (alleles, SNP, amino-acid polymorphisms) is reported. A dendrogram representing hierarchical clustering with the complete linkage of association *p*-values across the five MRI metrics is reported. We used turquoise color to encode metrics for gray volumes (GMV), violet color for the white matter volume (WMV) and orange color for the T2 lesion volume. Correlation between HLAGB and thalamic volume. Scatter plot of the HLAGB and thalamic volume (ThalV) with overlaid regression line. Each dot corresponds to a patient.

**Table 1 genes-13-02136-t001:** Demographic, clinical and MRI measures of the cohort.

Demographic and Clinical Features	
MS subjects (n)	334
Sex (females:males)	216:118
Age at MRI (years) (mean, range)	38.8 [18.1, 54.9]
Disease duration at MRI (years) (mean, range)	11.2 [0, 34.4]
Clinical course at MRI (RRMS; SPMS; PPMS)	253; 49; 32
Treated at time of MRI (y/n)	273/61
Treatment at MRI: 1st line; 2nd line; immunosuppressor; other; NA	169; 70; 21; 5; 8
Brain MRI metrics (mL)	
WMV (mean, range)	817.4 [553.3, 970.9]
GMV (mean, range)	689.4 [433.9, 875.5]
HippV (mean, range)	9.2 [5.3, 11.8]
ThalV (mean, range)	19.1 [11.9, 23.5]
T2LV (mean, range)	8.9 [0.04, 62.2]

The first-line group includes the following therapies: teriflunomide, dimetyhl fumarate, glatiramer acetate and interferons. The second-line group includes the following drugs: natalizumab and fingolimod. Abbreviations: MS = multiple sclerosis; RRMS = relapsing–remitting MS; SPMS = secondary–progressive MS; PPMS = primary–progressive MS; MRI = magnetic resonance imaging; WMV = white matter volume; GMV = gray matter volume; HippV = hippocampal volume; ThalV = thalamic volume; T2LV = T2 lesion volume.

**Table 2 genes-13-02136-t002:** Included covariates and HLAGB association statistics.

	Included Covariates						HLAGB Associations		
Brain MRI metrics	Age	Disease	Sex	Course	Treatment	Chip	β	95%CI	*p*
WMV	**100%**	**100%**	17%	56%	29%	33%	−2.825	[−10.831, 5.181]	0.488
GMV	**100%**	**100%**	19%	**100%**	35%	14%	−1.609	[−10.898, 7.680]	0.733
HippV	**100%**	**98%**	**100%**	**100%**	38%	28%	−0.142	[−0.291, 0.008]	0.064
ThalV	**100%**	**100%**	**77%**	**96%**	53%	32%	−0.299	[−0.595, −0.004]	0.047
T2LV	**100%**	**100%**	19%	**100%**	25%	42%	0.746	[−0.522, 2.014]	0.248

The percentage of times each covariate was selected in linear models across 1000 boostrapped samples is reported (bold: included, regular: not included). Abbreviations: HLAGB = HLA genetic burden; WMV = white matter volume; GMV = gray matter volume; HippV = hippocampal volume; ThalV = thalamic volume; T2LV = T2 lesion volume; CI = confidential interval.

**Table 3 genes-13-02136-t003:** HLAGB association statistics in relapsing–remitting and progressive patients (SPMS + PPMS).

	HLAGB Associations in Relapsing–Remitting Patients (n = 253)			HLAGB Associations in Progressive Patients (n = 81)		
Brain MRI metrics	β	95%CI	*p*	β	95%CI	*p*
WMV	−2.643	[−12.050, 6.763]	0.581	0.838	[−16.357, 18.034]	0.923
GMV	−0.853	[−11.126, 9.418]	0.871	−4.267	[−25.749, 17.216]	0.693
HippV	−0.123	[−0.287, 0.041]	0.141	−0.239	[−0.606, 0.128]	0.198
ThalV	−0.201	[−0.528, −0.126]	0.226	−0.673	[−1.356, 0.009]	0.053
T2LV	0.847	[−0.370, 2.066]	0.171	1.289	[−2.112, 4.689]	0.453

Abbreviations: HLAGB = HLA genetic burden; WMV = white matter volume; GMV = gray matter volume; HippV = hippocampal volume; ThalV = thalamic volume; T2LV = T2 lesion volume; CI = confidential interval.

## Data Availability

The datasets analyzed during the current study are not publicly available due to privacy restriction but are available from the corresponding author on reasonable request.

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
