# Peer review of "Exploring the Association of HLA Genetic Risk Burden on Thalamic and Hippocampal Atrophy in Multiple Sclerosis Patients"

_genes, 2022, doi:10.3390/genes13112136_

Round 1

Reviewer 1 Report

Santoro S et al. present original research focused on the association of cumulative HLA genetic burden and neuroimaging measures of brain atrophy in 334 Italian patients suffering from different clinical forms of multiple sclerosis. They found no correlation between HLA GB and volumetric white and grey matter measures. However, a weak association was described between HLA GB and thalamic volume, whereas a correlation with a tendency to significance was found between higher HLA GB and lower hippocampal volumes. Although the HLA associations with neuroimaging measures are a debated topic with contradictory results, these latter are new results worth publishing. The results are correctly presented, discussions are brief and logical, and conclusions are moderate and justified.

I have two formal proposals to improve the quality of this manuscript. First, Supplementary Figure 1. and its caption should be included in the manuscript, as Figure 2. in the Results section.

Second, the Materials and methods section should be structured into several subsections for a better understanding. With these improvements performed,

Author Response

  1. According to reviewer’s suggestion, we moved the heatmap of univariate HLA associations p-values from Supplementary Figure 1 to Figure 2. The caption of Figure 2 has been included in the manuscript.
  2. We agree with the reviewer that the Materials and Methods section needs to be better structured in subsections to improve its readability: we correspondingly divided it in four subsections, “Enrolled patients”, “Genotyping and imputation”, “Selected HLA loci” and “Statistical Analysis”.

Reviewer 2 Report

In the current manuscript, entitled” Exploring the association of HLA genetic risk burden on thalamic and hippocampal atrophy in Multiple Sclerosis patients”, Santoro et al., investigated the association between the cumulative HLA genetic burden (HLAGB) and brain atrophy metrics in MS patients. They analyzed an Italian cohort of 334 MS patients with imputed HLA alleles and cross-sectional volumetric measures of white matter (WM), gray matter (GM), hippocampus, thalamus and T2-hyperintense lesions. They found no effect of HLAGB on WM, GM and T2 lesion volumes. They revealed a marginal correlation between higher HLAGB and lower hippocampal volume and a nominal association between higher HLAGB and lower thalamic volume. They proposed a possible cumulative contribution of HLA risk loci on brain volumetric traits linked to clinical deficits in MS.

In general, this paper was well organized, and data was presented in a logistic way. I only have some minor points.

1. Sentence in line 30 and 34 are confusing in terms of GM. Please refine the line34.

2. In line 112-114, how the authors take these factors into account when they calculated the association between HLAGB and MRI metrics? Please provide detailed method for this part.

3. In line 214, the author claimed no association being identified in terms of gender, do the authors find any association in the context of disease course?

Author Response

  1. We thank the reviewer for highlighting this point in the Abstract: we modified it by explicitly referring the impact of HLAGB score to hippocampus and thalamus.
  2. We modified the paragraph in “Statistical analysis” subsection to improve readability, regarding the bootstrap procedure for the selection of factors to incorporate in the linear models.
  3. We thank the reviewer for the suggestion of stratifying the patients based on disease course; this analysis, although at the price of reduced sample size, is indeed of interest. We thus tested the association of HLAGB in patients with relapsing-remitting course and in those with progressive course, aggregating SPMS and PPMS. We interestingly found a nominal correlation between higher HLAGB and decreased lower thalamic volume (beta= -0.673, p= 0.053) in the progressive group, which thus seems to drive the association observed in the entire cohort. Details of results have been reported in the Results section and in Table 3. A comment regarding the obtained results has been included in the Discussion.